# Feasibility of Patient Navigation-Based Smoking Cessation Program in Cancer Patients

**DOI:** 10.3390/ijerph19074034

**Published:** 2022-03-29

**Authors:** Tongyao Fan, Jessica M. Yingst, Rebecca Bascom, Leonard Tuanquin, Susan Veldheer, Steven Branstetter, Jonathan Foulds, Joshua E. Muscat

**Affiliations:** 1Department of Public Health Sciences, College of Medicine, Penn State University, Hershey, PA 17033, USA; jyingst@phs.psu.edu (J.M.Y.); rbascom@pennstatehealth.psu.edu (R.B.); sveldheer@pennstatehealth.psu.edu (S.V.); jfoulds@psu.edu (J.F.); jmuscat@pennstatehealth.psu.edu (J.E.M.); 2Department of Medicine, College of Medicine, Penn State University, Hershey, PA 17033, USA; 3Department of Radiation Oncology, Penn State Milton S. Hershey Medical Center, Hershey, PA 17033, USA; ltuanquin@pennstatehealth.psu.edu; 4Department of Family and Community Medicine, College of Medicine, Penn State University, Hershey, PA 17033, USA; 5Department of Biobehavioral Health, Penn State University, University Park, PA 16802, USA; sab57@psu.edu

**Keywords:** cancer, smoking cessation treatment, patient navigation

## Abstract

Continued smoking after a cancer diagnosis is causally associated with increased risks of all-cause and cancer-specific mortality, and of smoking-related second primary cancers. Patient navigation provides individualized assistance to address barriers to smoking cessation treatment and represents a promising bridge to smoking cessation in persons with cancer who smoke cigarettes. We conducted a single-arm interventional cohort study of current smokers identified through prospective health record screening and recruited from Penn State Cancer Institute outpatient clinics. Consented participants received two telephone intervention sessions and gain-framed messaging-based smoking cessation educational materials designed for persons with cancer. The primary study outcome was the feasibility of the patient navigation-based intervention; the secondary outcome was the engagement in smoking cessation treatment at the two-month follow-up. Of 1168 unique screened Cancer Institute patients, 134 (11.5%) were identified as current cigarette smokers. Among 67 patients approached at outpatient clinics, 24 (35.8%) were interested in participating, 12 (17.9%) were enrolled, eight (11.9%) completed the intervention sessions and study assessments, and six engaged in smoking cessation treatment. The participants expressed satisfaction with the intervention sessions (median = 8.5, scale 0–10). The low recruitment rates preclude patient navigation as a feasible method for connecting cancer patients to smoking cessation treatment resources.

## 1. Introduction

Cigarette smoking is a well-established risk factor for more than 10 cancers, including lung, bladder, head and neck, liver, and esophageal cancers [1]. Overall, an estimated 20–30% of all cancer patients are current cigarette smokers at the time of their cancer diagnosis [2]. Among long-term cancer survivors, the prevalence of smoking ranges from 9% to 13% [2,3]. Continued smoking after cancer diagnosis is causally associated with an increased risk of both all-cause and cancer-specific mortality, and of smoking-related second primary cancers [1]. In addition, evidence suggests that continued smoking is strongly associated with an increased risk of cancer recurrence, a poor treatment response, and treatment-related toxic effects [1]. Continued smoking can also impact health care costs in cancer patients and lead to incremental costs due to the failure of first-line cancer treatment; compared with nonsmokers, the attributable costs are estimated as $10,700 per patient [4]. Cancer patients are able to experience multiple benefits by quitting cigarette smoking, including fewer treatment complications, the improved effectiveness of cancer treatments, improved survival, a reduced risk of future second primary tumors, and improved physical and psychological functioning [5,6,7,8].

Smoking cessation treatment has been recommended as an essential component of cancer care by leading national organizations [5,9,10]. Since 2017, the National Cancer Institute (NCI) has funded 52 NCI-designated cancer centers to implement science-based smoking cessation treatment into clinical practice [11,12]. However, such treatments are not frequently provided to cancer patients as part of their cancer care [5,13,14,15]. While the Clinical Practice Guidelines recommend that oncology clinicians provide smoking cessation treatment for cancer patients [16], most oncology clinicians do not routinely assist, with cited reasons including a lack of time, training, or resources [17,18,19,20].

Patient navigation is a patient-centered health care service delivery model that addresses the barriers to quality standard care by providing individualized assistance to patients, survivors, and families [21]. A patient navigator often builds a one-on-one relationship with a patient and provides services including support, education, and referral to resources, thus addressing individual barriers and needs across the continuum of cancer care [22]. Patient navigation programs have been proven to improve access to health screenings in cancer patients and to decrease the time from an abnormal finding to a diagnostic resolution [22,23]. In addition, patient navigation is also considered as a promising and acceptable strategy to link patients with community resources [24]. In one study, patients with a patient navigator reported a significantly higher score for their learning of available community resources compared to a control group who lacked a patient navigator [25]. Most previous publications have focused on assessing the effectiveness of patient navigation for screening (50%) and diagnosis (27%) along the continuum of cancer care, and to a lesser extent cancer outcome [22,23]. By providing individualized assistance and addressing barriers, patient navigation represents a promising bridge between the cancer patient and potentially accessible smoking cessation treatment resources. 

In cancer care, smoking cessation treatment has been referred to as the “fourth pillar of cancer care”, along with surgery, radiotherapy, and chemotherapy [26]. Strategies must also be developed in cancer centers with limited funding to promote the connection of smokers with cancer to the available smoking cessation treatment resources. Based on the current evidence, we believe that a promising approach to improve smoking cessation treatment is to prospectively provide navigation services to persons with cancer who are current cigarette smokers. The purpose of this paper is to study the use of patient navigation to promote the connection to smoking cessation treatment resources in cancer patients who are active cigarette smokers and discuss barriers.

## 2. Materials and Methods

A single-arm cohort study design (Figure 1) was used to test the feasibility of the patient navigation-based intervention to promote the connection to smoking cessation treatment resources in cancer patients. During the study period, all participants received gain-framed smoking cessation educational materials and two intervention sessions (by phone) separated by one month and completed two study assessments. All study materials and procedures were approved by the Penn State Cancer Institute (PSCI) Protocol Review Committee (PRC) and the Penn State Institutional Review Board (IRB) under study ID# STUDY00017696. The study was also registered on the ClinicalTrials.gov website (identifier: NCT04972916). The study visits and timing are outlined below in Figure 1. 

### 2.1. Study Setting and Stakeholder Engagement

The current study was conducted at the PSCI, a cancer center located on the campus of the Penn State Milton S. Hershey Medical Center. We collaborated with the PSCI head and neck cancer and thoracic cancer disease teams to identify and recruit cancer patients who were also current cigarette smokers. These multidisciplinary teams consisted of surgeons, radiation oncologists, medical oncologists, and pathologists to guide patients through every step of their care. Three medical oncologists, four head and neck cancer surgeons, and nine radiation oncologists agreed to participate in this study. All outpatient clinics used a common electronic health record (EHR) system called Cerner CareConnect/PowerChart to support healthcare services. For each clinical appointment, all clinics had medical assistants who screened for cigarette smoking as part of patient intake, recording the information in the EHR system. The multidisciplinary team clinicians individually decided whether to further assess and provide smoking cessation treatment support or defer it to primary care providers.

### 2.2. Study Population and Recruitment 

#### 2.2.1. Inclusion Criteria

The inclusion criteria for the study participants included cancer patients who (1) were ≥ 18 years old; (2) smoked at least one cigarette in the last 30 days; (3) read and spoke English; and (4) were able and willing to participate in the study protocol and provide consent. Cancer patients were invited to participate regardless of their cancer type, stage, or time since diagnosis. We also included patients with multiple cancer diagnoses or a previous history of cancer. Patients who were not ready to quit but were interested in reducing their smoking or in obtaining more information about quitting were also eligible. We excluded patients who were aged < 18 years old, unable to read and speak English, receiving radiation therapy, or actively using evidence-based smoking cessation treatment (pharmacotherapy or behavioral counseling) at the time of recruitment.

#### 2.2.2. Participant Identification

We identified potential study participants by reviewing their most recent cigarette smoking status in the EHR on a weekly basis for each clinic. The cigarette smoking status information could be found in the “Social History”, “Outpatient Visit Summary”, or “Outpatient visit note” sections. We reviewed all three parts for each individual patient and identified patients by their smoking status, collected at their most recent encounter. The cigarette smoking status was identified by the following question from the Adult Admission Health Habits Assessment form in the EHR: “Have you ever been a cigarette smoker or do you currently smoke?” Possible responses included:Never smoked cigarettes;Current everyday light smoker (less than 10 cigarettes per day);Current everyday heavy smoker (greater than 10 cigarettes per day);Current someday light smoker (less than 10 cigarettes per day);Current someday heavy smoker (greater than 10 cigarettes per day);Former smoker, quit in the last 30 days;Former smoker, quit within 31 days–1 year;Former smoker, quit greater than 1 year ago;Smoker, current status unknown;Unknown if ever smoked;Patient refused to answer.

“Current smoker” included the current everyday light/heavy smokers, current someday light/heavy smokers, and former smokers who quit in the last 30 days; all were potentially eligible for this study. A “former smoker” was defined as someone who had quit smoking for greater than 30 days. Patients who were recorded as “smokers, current status unknown”, “unknown if ever smoked”, or “patients refused to answer” were categorized as cigarette smoking status unknown. Every week, the researcher developed an eligible patient list and sent it to the nurse coordinators before clinical days to further confirm their eligibility and to verify the patient’s schedule.

#### 2.2.3. Participant Recruitment

The participant recruitment lasted 3 months (15 June 2021–16 September 2021) at the medical oncology clinic, 2.5 months (15 June 2021–16 September 2021) at the head and neck surgery clinic, and 1 month (17 August 2021–16 September 2021) at the radiation oncology clinic. At the suggestion of the radiation oncologist, we excluded patients who were actively receiving radiation treatments from recruitment. 

On clinical days, the researcher approached eligible patients at their visits after they were seen by the medical assistant. The researcher followed the IRB-approved script to provide a brief self-introduction, introduce the project, and ask if the potential participants were interested in the study. To those who were interested, the researcher provided a copy of a summary explanation of the research, confirmed their phone number and/or email address, and scheduled a time within one week to call and obtain consent. The researcher called interested patients based on their scheduled time. Up to 3 attempts to contact the patient were made, separated by 1 week. A text message, email, or voice message was sent or left at the third phone call. Upon successfully reaching the participant, the researcher reviewed the summary explanation of the research and further screened their eligibility (e.g., not using any smoking cessation treatment at the time of the call). Those who were eligible and provided verbal consent were considered as successfully enrolled.

At each clinical visit, the participants’ cigarette smoking status was collected as part of clinical intake. We collected this information for all screened patients to compare their self-reported cigarette smoking status.

To maximize enrollment, we (1) provided incentives for participating ($30 gift card); (2) confirmed phone numbers and email addresses with patients at the clinics; (3) obtained a “best date/time to call” from the participants; (4) made three attempts on different days to reach the patients; (5) used multiple channels (e.g., email, text message, and voice message) to contact patients; and (6) re-approached interested patients in the clinic who were not able to be reached by phone.

### 2.3. Baseline Assessment

The baseline survey was conducted by phone at the time of or within one week of the consent call. We first collected their baseline EHR data including cancer types, a diagnosis of tobacco use disorder (Diagnostic and Statistical Manual of Mental Disorders diagnosis), and cessation medication prescriptions. The baseline assessment contained questions on demographics, cigarette smoking behaviors, nicotine dependence, cessation history, the use of other tobacco products, the importance of and confidence in quitting, and the intention to quit.

Demographics: Participants answered a range of questions related to their age, ethnicity, gender, and education. 

Smoking history: Smoking history including smoking years, use of other tobacco products, and previous quitting attempts and the utilization of smoking cessation treatment were collected. 

Nicotine dependence: We measured the level of nicotine dependence with the Fagerstrom Test for Nicotine Dependence (FTND) [27]. The FTND is the most widely used self-report of nicotine dependence and has consistently demonstrated good psychometric properties across a range of populations of smokers.

Stage of change: The stage of change regarding smoking cessation was assessed by the commonly-used measure introduced by Prochaska and colleagues to classify smokers into the following categories: pre-contemplation (not planning to quit), contemplation (planning to quit within the next 6 months), or preparation (planning to quit within the next month) [28]. 

Importance of and confidence in quitting: As a direct measure of the self-reported importance of quitting, we asked participants to rate, on a 10-point Likert scale, how important it is stop smoking now (1 = not important; 10 = very important). To assess confidence in quitting, we asked participants to rate how confident they were that they would be able to stop smoking (1 = not at all confident; 10 = very confident). These face-valid questions have been utilized in numerous studies of smoking and smoking cessation and are predictive of cessation success or failure [29,30,31].

### 2.4. Study Intervention 

#### 2.4.1. Patient Navigator

The navigator was a nurse practitioner at the Penn State Hershey Medical Center specializing in hematologic and pulmonary disease. The navigator had extensive clinical experience working with cancer patients and smoking patients. The phone-based intervention sessions were scheduled every Monday afternoon (1:00 p.m.–3:30 p.m.) and Friday morning (9:30 a.m.–12:00 p.m.) during the study period. The navigator was also flexible and able to conduct intervention sessions based on participants’ preferences if these two sections did not work for them.

#### 2.4.2. Smoking Cessation Education Material

We chose gain-framed messages to design our smoking cessation education material specifically for cancer patients. The gain-framed message refers to the emphasis of a message on the benefits of an intervention or behavior [32]. Previous studies have shown that gain-framed messages seem to produce positive effects on adult patients who smoke cigarettes [33]. The education material focused on discussing the benefits of quitting and reducing cigarette smoking after cancer diagnosis as well as tobacco harm reduction messages. 

#### 2.4.3. First Navigation Session

The first patient navigation intervention was conducted by phone within 2 weeks after the baseline survey. The nurse navigator called enrolled participants based on their day/time scheduled at the baseline survey. For those who did not answer, up to 3 attempts to contact the patient were made, separated by 1 week. Upon successfully reaching the participant, the navigator followed the IRB-approved intervention script to conduct the intervention session, which was based on the 5As model [16]. The smoking cessation education brochure was emailed or mailed to the participants after the intervention session.

(1)Asked the patients about current cigarette use.(2)Advised patients about the risks of continued smoking after cancer diagnosis and the benefits of quitting. For those who just wanted to reduce but not stop smoking entirely, the benefits of reducing were advised. The navigator also discussed nicotine replacement therapy (NRT) or other nicotine products upon the participants’ requests.(3)Assessed patients’ intention to quit.(4)Assisted the patients with smoking cessation by referring them to available smoking cessation treatment resources, including PA Free Quitline or Penn State Smoking Cessation Clinic, through the EHR. At the same time, the navigator assisted patients in identifying potential barriers to smoking cessation and referred them to relevant resources if they needed.The Penn State Health Smoking Cessation Clinic, started in July 2018, is a smoking treatment program that involves one-on-one, 30-minute sessions to assess smokers’ nicotine dependence. During the treatment session, they discuss the smokers’ smoking routine, review strategies, and evaluate coping skills. Then, they develop a tailored plan that is specific to each individual smoker’s lifestyle and the lifestyle modifications that will help them quit. The team is also able to prescribe medications to assist in controlling nicotine cravings. Anyone who is currently smoking and has a desire to quit is eligible for this program.The PA Free Quitline is a free phone-based counseling service that addresses cessation needs and impacts statewide tobacco use. It provides a series of up to five phone-based cessation counseling sessions, educational material, and NRT qualified callers [34].(5)Arranged one-month follow-up calls. At the end of first intervention interview, the date and time for the second patient navigation intervention session were scheduled.

#### 2.4.4. Second Navigation Session

The second intervention interview was conducted by phone and happened one month after the first patient navigation session. The navigator assessed the participants’ use of smoking cessation treatment. Among those who had not used any smoking cessation treatment (counseling or medication) since the first session, the navigator worked with the participant to explore barriers and motivated them to use smoking cessation treatment. Among those who had used smoking cessation treatment, the navigator encouraged them to continue. At the end of the second intervention interview, all participants were reminded that the final assessment would happen one month after the second intervention interview. They were asked their preferred format of final assessment (by email or phone) and for a date/time that worked best for them.

#### 2.4.5. Incentive

Participants who completed the intervention phase received up to a total of $30, paid via Greenphire ClinCard: $10 for each intervention session and $10 for the final assessment. The ClinCard was provided to participants at their return clinic visit after the completion of their first intervention session. For participants who did not have an upcoming clinical visit, the card was mailed to the address they provided.

### 2.5. Two-Month Follow-Up Assessment

The final survey was conducted one month after the second patient navigation session. Study subjects were emailed a survey link or called to complete a survey questionnaire that examined current cigarette smoking status, cigarette smoking behaviors, nicotine dependence, perceived risks of smoking, engagement in smoking cessation treatment, and satisfaction with the patient navigation intervention sessions.

Smoking-related questions: First, participants were asked about their current smoking status. Those who were self-reported current everyday or someday smokers at the time of the final assessment were assessed for nicotine dependence, the importance of and confidence in quitting, and stage of change using the same sets of questions from the baseline questionnaire.

Use of smoking cessation treatment: Participants’ self-reported use of behavioral counseling services or cessation medications during the study period was collected.

Satisfaction: To assess the participants’ satisfaction with the intervention sessions, we asked them to use a Likert scale to rate how satisfied they were with the intervention sessions they received (1 = not at all satisfied; 10 = very satisfied). We also asked open-ended question to assess the most helpful aspects of the program.

### 2.6. Data Collection and Statistical Analysis

All survey data were collected and stored in the Penn State Research Electronic Data Capture (REDCap), which is a secure, web-based application designed to support data capture for research studies in over 600 U.S. academic institutions [35]. The patient navigator’s intervention notes were summarized and entered into REDCap by the researcher. 

We evaluated the feasibility of the suggested approach by tracking the recruitment process. We also examined descriptive statistics from approached patients and compared interested and non-interested patients to determine whether there was any significant difference between the two groups. For categorical variables, numbers and percentages were reported, and for continuous variables, we examined means and standard deviations. Chi-square tests were used for categorical variables and t-tests were used for continuous variables. An exploratory analysis was conducted to determine whether the patient navigation intervention could result in changes in smoking behaviors at the 2-month follow-up compared to baseline. The participants’ engagement in smoking cessation treatment and satisfaction with intervention sessions were also measured at follow-up to evaluate the feasibility of the program. Engagement in treatment was defined as the completion of at least one smoking cessation counseling session (e.g., at the Penn State Smoking Cessation Clinic or Quitline) or the use of at least one Food and Drug Administration (FDA) approved smoking cessation medication for at least 1 day during the study period. 

This pilot study consisted primarily of descriptive analyses and was not powered to conduct hypothesis testing for significant findings. Significance was set at a level of 0.05 for all analyses. All analyses were performed using the SAS statistical software (SAS Institute Inc., Cary, NC, USA) and REDCap.

## 3. Results

### 3.1. Participant Recruitment Flow

A total of 1972 adult patients’ EHRs were screened for eligibility, of which 1168 were unique records (Figure 2). The mean age of the 1168 unique patients was 63.3 years (SD = 14.1). One-third (407, 34.8%) of the patients had clinical appointments scheduled at medical oncology clinics, 542 (46.4%) at head and neck surgery clinics, and 219 (18.8%) at radiation oncology clinics. Among 984 (84.2%) patients who had cigarette use information available in their medical record at the time of screening, 134 (13.6%) were current smokers, including 58 current everyday heavy smokers, 55 current everyday light smokers, seven current someday light smokers, and 14 former smokers who quit in the last 30 days. In addition, 34.4% (N = 339) were former smokers (quit > 30 days) and 51.9% (N = 511) were never smokers.

We further excluded 40 patients who were determined not to have cancer after their clinic visits, and two patients who did not speak English (Figure 2). Among these eligible patients (N = 92), the researcher approached 60 (65.2%) patients at the clinics. Twelve (13.0%) patients were not approached because of appointment cancellations or no-shows; 12 (13.0%) patients were missed because of scheduling conflicts; and five (5.4%) patients who were identified as non-current smokers at their clinical intake were excluded. Three patients were not recommended by the nurse due to their clinical status at the time of recruitment: two patients had advanced disease and emotional breakdown at clinics and the other patient suffered from medication side effects. 

We also tracked and recorded cigarette smoking status from the clinical intake at the end of each recruitment week. Among 1168 patients, 86.8% (N = 1020) were screened for cigarette smoking as part of their clinical intake. Of these, 877 patients had cigarette smoking information available at two time points, and 745 (84.9%) patients had a consistent cigarette smoking status. Among those who had inconsistent cigarette smoking information, 33 out of 39 never smokers in pre-screening were recorded as former smokers at clinic intake, and 69 out of 76 former smokers were screened as never smokers (Table 1). We further identified 13 current smokers who were identified as never smoker, former smoker, or unknown at the time of pre-screening. The researcher was able to approach seven out of these thirteen patients at their return clinical visits.

Among the total of 67 patients that were approached, 38 (56.7%) patients were not interested, two patients (3.0%) were unsure, three (4.5%) patients had quit smoking > 30 days ago, and 24 (35.8%) patients were interested in participating. The researcher was only able to reach 13 of the 24 interested patients by phone, and three declined to participate at the time of the phone call. Among 11 patients who were not able to be reached by phone (three calls) after the initial approach, the researcher re-approached four patients at their return clinic visits and enrolled two patients. A final total of 12 participants were enrolled in this study.

### 3.2. Comparison of Interested and Non-Interested Patients

The mean age of the approached patients was 62.0 years (SD = 6.9); 37.3% were married (Table 2). The majority of the approached patients were White (95.5%) and male (64.2%). Forty-two (62.7%) patients were diagnosed with tobacco use disorder in their EHR, and 10 (14.9%) patients had cessation medication prescriptions recorded in their EHR medication list. There was no significant difference between the interested and non-interested groups.

### 3.3. Study Participants

#### 3.3.1. Baseline Characteristics

Among the 12 participants who consented to enroll in this study, 10 (83.3%) completed the baseline survey. The other two patients indicated that they would prefer to complete the baseline survey by email, but never responded to the REDCap link sent to them. The mean age of the participants was 58.4 years; 100% were White (Table 3). Most participants were male (70%), single (60%), diagnosed with lung cancer (80%), and had a high school or lower education level (60%). The participants had been smoking for an average of 44.7 (median = 46.5) years and smoked 13.3 (median = 11) cigarettes per day. The average FTND nicotine dependence level was low (mean = 3.9). Half smoked menthol-flavored cigarettes; 40% had ever used electronic cigarettes (20%), cigars (10%), or pipes (10%); and 20% had used electronic cigarettes (10%) or cigars (10%) in the last 30 days. All participants had tried to quit smoking in the past and had been advised by their health care providers to quit since their cancer diagnosis. The participants reported an average of 5.2 quit attempts made in the last 12 months. For the majority (70%) of participants, the longest time they were able to stay quit was less than one month. All participants had used NRT in the past; however, only 10% had ever used behavioral counseling services. The median score of the rated importance of quitting smoking now was 10, but the median confidence score was only 5. The majority of participants (70%) were at the preparation or contemplation stage of smoking cessation. 

#### 3.3.2. Referral to Smoking Cessation Treatment

The navigator was able to reach nine (90%) out of ten participants to conduct the first intervention. An average of 1.1 phone calls were made to in order to reach the participants. Seven participants were referred to smoking cessation treatment resources: one was referred to Quitline, five were referred to the Penn State Smoking Cessation Clinic, and one was connected with resources from the American Lung Association. Two participants were not ready for any treatment at the time of the first intervention session. 

One participant was not ready to make any changes and refused the second intervention call from the navigator. Eight participants completed the second intervention sessions with the navigator. After the second intervention session, one additional patient was referred to Quitline. All eight participants who completed two intervention sessions were referred to smoking cessation treatment recourses by the navigator.

### 3.4. Two-Month Follow Up

Eight participants completed the two-month follow-up assessment. Two (25%) participants had already quit smoking and six (75%) were still daily smokers. A total of six (75%) participants engaged in smoking cessation treatment: two used NRT only and four used counseling services and NRT during the study period (Table 4). 

Compared to baseline, participants reported fewer cigarettes per day (CPD; median: 11 vs. 6.5), more advanced stages (at or beyond contemplation) for smoking cessation (70% vs. 100%), and higher confidence in quitting smoking (median: 5 vs. 6) at the two-month follow-up. Overall, they were satisfied with the intervention sessions (median = 8.5), and in the words of one participant: ”She (the navigator) listened and understood, and made me feel I am not alone.” Two participants identified barriers to the use of smoking cessation treatment at their follow-up assessment, including “insurance does not pay for treatments” and “NRT is not working anymore”.

## 4. Discussion

The current study sought to address smoking cessation treatment in cancer patients by first utilizing a patient navigation model to guide the development of a smoking cessation program. This paper aimed to provide preliminary data on using patient navigation to promote a connection to smoking cessation treatment in cancer patients. 

In total, 35.8% of the approached patients were interested in participating and 17.9% were enrolled in current study. Patients who were not interested in participating stated that they had too much going on at the time of recruitment and were not ready to quit; they did not believe this would work on them; or they planned to quit by themselves. Participant recruitment for smoking pharmacotherapy trials or physician-initiated interventions has been shown to range from 16.5–84% [36,37,38]. The NCI Cancer Moonshot Funded Cancer Center Cessation Initiative reported participation rates of 3.4–87.3% [39]. By the end of 2018, 11 NCI-designated cancer centers implemented optional EHR referral, and nine centers used automatic EHR referral to increase the reach of smoking cessation treatment programs in cancer patients [39]. Using an EHR-based referral approach could increase the number of patient referrals. However, the percentage of those referrals who engage in treatment could range from 17% [40] to 43.1% [41]. Some of these EHR-based programs have relied on clinicians to advise patients to use smoking cessation treatment, which is not always feasible for clinicians who do have enough training, time, or resources. An important aspect for implementing a smoking cessation program in cancer patients is to secure provider buy-in. We successfully engaged nurses and clinicians by building a relationship and consensus with disease team leadership, enhancing compatibility with clinical care workflow, and reducing their burden. The removal of the burden of providing smoking cessation treatment support from clinicians could promote the adoption of smoking cessation programs into oncology clinical settings.

In our screened patients, we found that 11.3% had a different cigarette smoking status at their clinical intake compared to our prospective EHR review. This may suggest that patients self-reported their cigarette smoking status differently at different clinical visits. In addition, some cancer patients may stop smoking cigarettes temporarily because of cancer treatments or hospitalization, and relapse afterwards [42,43]. For those conducting smoking cessation projects in cancer patients and using similar methodology to identify eligible patients, our findings suggest that future studies should continue to screen returning patients (not just first-visit clinic patients) and reach out to smoking patients on multiple occasions over time. In addition, 24 out of 92 eligible patients were not approached because of a change in their clinical schedule or scheduling conflicts. Future studies may consider using remote assessments to increase recruitment rates.

We did not find any significant differences in demographic or clinical variables between patients who were interested in participating and those who were not. One previous study showed that willingness to participate in a cessation program was similar across a broad range of sociodemographic factors and tumor characteristics [44], while other studies have suggested that patients who are male, White, married, diagnosed with non-smoking related cancer, or with a shorter time since diagnosis are more likely to engage in cessation programs [45,46,47]. Our results also revealed that 62% of the approached patients had a tobacco use disorder diagnosis in their EHRs, and patients who were not interested in participating had an even higher proportion of tobacco use diagnosis. Future research might be able to investigate the reasons why patients who have a tobacco-related diagnosis may not be interested in smoking cessation programs.

This study also introduced the concepts of harm reduction in the cancer patient population, both through educational materials and also by assessing the past usage of alternative tobacco products. Our results show that 40% of the participants had ever used other tobacco products, with 10% having used an electronic cigarette in the past 30 days. Other studies have found that 3–25% of patients with cancer are current users of electronic cigarettes [48,49]. Future research could further explore whether smoking cessation programs should consider counselling to encourage the use and success of alternative tobacco products in patients who are unable to quit smoking cigarettes.

One barrier we identified was that participants who were referred to Quitline had not received any phone call from Quitline at the time of the follow-up assessment. We checked with the Quitline administrator and confirmed that the referrals were successfully made. It is possible that participants may have missed the calls from Quitline or could not recognize their numbers. For future research, understanding the referral process for outside resources and following up with referred patients periodically might help to increase their use of smoking cessation treatment. 

The main limitations of our study include the small sample size and short study period. We were not able to provide translation services for non-English speaking patients, although PA Quitline provides Spanish-based counseling. In addition, we evaluated smoking behaviors by self-reporting; future research should biochemically verify smoking status. We did not measure 7-day or 30-day point prevalence smoking abstinence in the 2-month follow-up assessment. Our study was conducted in one health care system and the majority of the participants were White, potentially limiting the generalizability of our results to other health care settings and racial groups. This pilot study had research staff involved in the process of identifying and assessing eligible patients; this could be a barrier for future implementation. 

## 5. Conclusions

The small number of patients enrolled in this pilot study preclude the recommendation of patient navigation to facilitate smoking cessation in cancer patients using the current methods. The potential for navigation might still be considered with additional considerations. Our results demonstrated the feasibility of this program with regard to engaging institutional, clinician, and clinical staff buy-in. Our population-based approach with 100% screening of clinic attendees provided accurate data on cancer patients’ smoking status and overall recruitment efficiency. Our detailed description of the steps of the patient navigation process identifies steps for improving participant recruitment. Future research may need to have more dedicated funding and resources to allow for more intensive efforts, including using remote assessments, constantly screening returned patients’ cigarette smoking status, reaching out to smoking patients on multiple occasions over time, consenting and distributing assessments in person, and re-approaching patients who expressed interest but were not reached, to improve smoking cessation treatment support for cancer patients. Further considerations for whether patient navigation is feasible for cancer treatment centers should examine the financial costs of a sustained program. Our study was conducted under COVID-19 safety protocols that restricted the time and space opportunities for interacting with patients. Mask wearing is another potential psychological barrier for engagement. Success rates may potentially be better under non-COVID-19 hospital protocols. 

## Figures and Tables

**Figure 1 ijerph-19-04034-f001:**
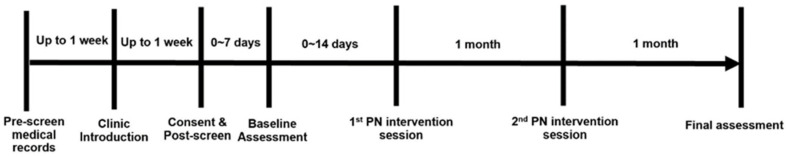
Study design.

**Figure 2 ijerph-19-04034-f002:**
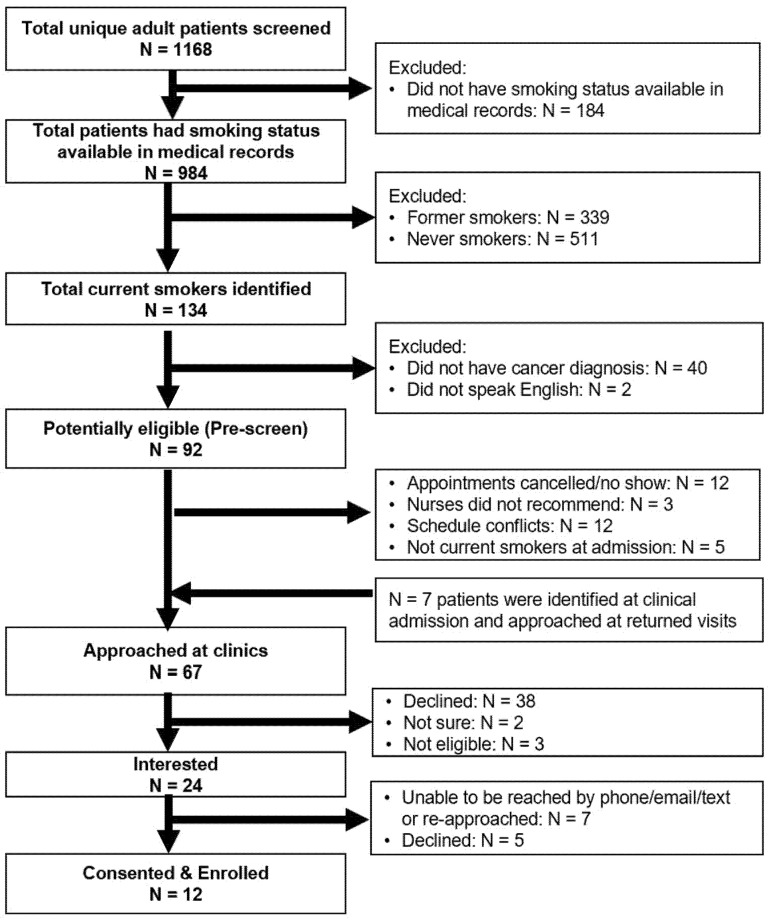
Recruitment diagram.

**Table 1 ijerph-19-04034-t001:** Inconsistent cigarette smoking status between EHR review and clinical intake.

EHR-Screen Cigarette Smoking Status, N	Clinical Intake Cigarette Smoking Status, N				
	Never	Current	Former	Unknown	Total
Never	0	6	33	0	39
Current	8	0	5	0	13
Former	69	5	0	2	76
Unknown	1	2	1	0	4
Total	78	13	39	2	132

**Table 2 ijerph-19-04034-t002:** Comparison of interested and non-interested patients.

Variables	Total N = 67	Not Interested N = 43	Interested N = 24	*p*-Value
Age, mean (SD)	62.0 (6.9)	62.8 (6.7)	60.6 (7.2)	0.21
Gender, n (%)				
Male	43 (64.2)	29 (67.4)	14 (58.3)	0.46
Female	24 (35.8)	14 (32.6)	10 (41.7)	
Race, n (%)				0.69
White	64 (95.5)	41 (95.4)	23 (95.8)	
Black or African American	2 (3.0)	1 (2.3)	1 (4.2)	
Multiple	1 (1.5)	1 (2.3)	0 (0)	
Marital status, n (%)				0.14
Married	25 (37.3)	16 (37.2)	9 (37.5)	
Divorced	13 (19.4)	11 (25.6)	2 (8.3)	
Separated	2 (3.0)	2 (4.7)	0 (0)	
Single	20 (29.9)	9 (20.9)	11 (45.8)	
Widowed	7 (10.4)	5 (11.6)	2 (8.3)	
Cigarette smoking status, n (%)				0.55
Current everyday heavy smoker	26 (38.8)	15 (34.9)	11 (45.8)	
Current everyday light smoker	30 (44.8)	19 (44.2)	11 (45.8)	
Current someday light smoker	4 (6.0)	3 (7.0)	1 (4.2)	
Former smoker, quit in the last 30 days	7 (10.4)	6 (14.0)	1 (4.2)	
Diagnosed of tobacco use disorder, n (%)				0.11
Yes	42 (62.7)	30 (69.8)	12 (50.0)	
No	25 (37.3)	13 (30.2)	12 (50.0)	
Cessation medication prescription, n (%)				0.08
Yes	10 (14.9)	4 (9.3)	6 (25.0)	
No	57 (85.1)	39 (90.7)	18 (75.0)	

**Table 3 ijerph-19-04034-t003:** Baseline characteristics.

Variables	Total N = 10
Age, mean (SD)	58.4 (58.5)
Male, n (%)	7 (70)
White, n (%)	10 (100)
Marital status, n (%)	
Married	3 (30.0)
Divorced	1 (10.0)
Single	6 (60.0)
Education, n (%)	
Less than high school	5 (50.0)
High school graduate	1 (10.0)
Some college/no degree	3 (30.0)
Associate degree	1 (10.0)
Cancer type, n (%)	
Lung cancer	8 (80.0)
Head and neck cancer	2 (20.0)
Years of smoking, mean (median)	44.7 (46.5)
Cigarettes per day, mean (median)	13.3 (11.0)
Smoke menthol cigarettes, n (%)	5 (50.0)
Had ever used other tobacco products, n (%)	4 (40.0)
Electronic cigarettes	2 (20.0)
Cigars	1 (10.0)
Pipes	1 (10.0)
FTND, mean (median)	3.9 (4)
Quit attempts in last 12 months, mean (median)	5.2 (2)
Longest time staying quit, n (%)	
Less than 1 week	2 (20.0)
1 week to 1 month	5 (50.0)
1 to 6 months	2 (20.0)
6–12 months	1 (10.0)
Used cessation treatment in the past, n (%)	
NRT	10 (100.0)
Medications	6 (60.0)
Behavioral counseling	1 (10.0)
Stage of change, n (%)	
Quit within the next month (preparation)	5 (50.0)
Quit within the next six months (contemplation)	2 (20.0)
Quit someday, but not next six months (precontemplation)	3 (30.0)
Not interested in quitting	0 (0.0)
Around smokers most of time, n (%)	5 (50.0)
Importance of stopping smoking, mean (median)	9.7 (10.0)
Confidence in stopping smoking, mean (median)	5.3 (5.0)

**Table 4 ijerph-19-04034-t004:** Two-month follow-up.

	Baseline N = 10	Two-Month Follow up N = 8
CPD, median	11	6.5
Importance of stopping smoking (1–10), median	10	10
Confidence in stopping smoking (1–10), median	5	6
Stage of change, n (%)		
Already quit (action)	-	2 (25)
Quit within the next month (preparation)	5 (50)	3 (37.5)
Quit within the next six months (contemplation)	2 (20)	3 (37.5)
Quit someday, but not next six months (precontemplation)	3 (30)	0 (0)
Engage in smoking cessation treatment, n (%)		
Behavioral counseling and NRT	-	4 (50)
NRT	-	2 (25)
Satisfaction with intervention sessions (1–10), median	-	8.5

## Data Availability

The data presented in this study are available on request from the corresponding author.

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
