# Peer review of "Feasibility of Patient Navigation-Based Smoking Cessation Program in Cancer Patients"

_ijerph, 2022, doi:10.3390/ijerph19074034_

Round 1
Reviewer 1 Report
The high quality literature review and excellent description of methocology and experience, in combination with the admission that low recruitment rates preclude patient navigation as a feasible modality for smoking cesstion in cancer patients make this study of both great interest and value as a contribution to the literature.
(what follows is reviewer opinion not recommended for modification of the paper unless the authors desire to do so) -- The underlying problem is lack of a smoking cessation intervention that these patients have not already tried. Since they have tried and failed before using counselling and NRT's, they see no point in trying again. If and when tobacco harm reduction or other interventions are approved for use, then patient navigagion utilizing more attractive and more effective modalities may prove to be feasible.
Reviewer 2 Report
Thank you so much for being responsive, and congratulations on conducting this study given how challenging it is to conduct studies with the cancer patient population.
This manuscript is a resubmission of an earlier submission. The following is a list of the peer review reports and author responses from that submission.
Round 1
Reviewer 1 Report
The present study analyzes feasibility and patient satisfaction of an intervention that used navigators to engage cancer patients who smoked at least once cigarette in the past month with smoking cessation services. The study also aims to provide preliminary data on effectiveness. I really enjoyed reading this manuscript and certainly appreciate the authors’ effort recruiting this challenging population. Smoking rates are high among the cancer patient population and smoking cessation trials have failed to find a significant intervention effect. Thus, this is an important study that fills a gap in the literature by addressing engagement with smoking cessation treatment. Although I do believe this study provides important and meaningful information, I believe that authors go beyond what can be concluded from the data provided by a pilot study. I am specifically concerned with the fact that authors try to assess preliminary effectiveness with a study that is clearly underpowered and without having a control group. Below are some other comments and suggestions:
- In the Introduction the authors indicate that reducing tobacco use can be beneficial for cancer patients (page 2, lines 52-54). I am not really sure there is literature to support this statement. The three studies cited measured smoking rates at the time of diagnosis and compared light versus heavy smokers, but they did not assess whether patients who reduced their smoking experienced any benefits. I do believe smoking reduction might be beneficial towards the final goal of complete smoking abstinence. I suggest removing the parts of the introduction dedicated to harm reduction, since there is not enough literature to support this and it is not the aim of the study.
- Pilot studies are usually not powered to test efficacy or effectiveness (https://www.nccih.nih.gov/grants/pilot-studies-common-uses-and-misuses). In addition, in this specific study there is no control group, so I think effectiveness cannot be tested.
- Assessment of feasibility is always complicated and I wonder if authors had established any benchmarks a priori or if they could compare recruitment rates in this study with other similar studies to support the conclusion that the study is indeed feasible. I completely understand the challenges of recruiting cancer patients who smoke to participate in trials, but it seems optimistic to conclude that the study is feasible when only 12 patients consented. How long would it take to recruit a sample for a fully powered trial? Authors may want to consider explaining lessons learned from this study to modify recruitment strategies that would increase accrual rates in a future study.
- Did patients read and provide feedback on the educational materials provided?
- Having patient navigators to establish a connection of patients who smoke with smoking cessation services seems like a great idea and it can take that burden from health care providers. However, there was research staff involved in the process of identifying and assessing patients who smoke. This could be a barrier for future implementation and maybe this should be added as a study limitation.
- Please specify in the inclusion/exclusion criteria what “actively using evidence-based tobacco treatment at the time of recruitment” means. Is this referring to pharmacotherapy or does it also include behavioral counseling?
- Were there any inclusion/exclusion criteria related to cancer type, stage, or time since diagnosis? Could patients have multiple cancer diagnosis or previous history of cancer?
- Why did recruitment last differently in each clinic?
- How did the navigator made the referral to smoking cessation services?
- Since patients actively receiving radiation treatments were excluded, please consider adding this to the exclusion criteria.
- When were the incentives provided?
- Could you please clarify what does “diagnosis of tobacco use disorder” mean in this context? Is it DSM criteria for tobacco use disorder or does it refer to a smoking-related cancer? If the latter, please specify what types of cancers are considered smoking-related.
- Authors indicate that importance and confidence in quitting have been used in numerous studies and are predictors of smoking outcomes, but they do not support this with citations (page 5, lines 230-232). Please provide references.
- Please indicate if the sessions conducted by the navigator were in person or over the phone (page 5, lines 237-239).
- Why did the first session with the navigator did not happen until two weeks after baseline? I wonder if some patients would have been smoke free for over 30 days or already engaged with smoking cessation services by the first call with the navigator.
- Authors may consider restructuring the Materials and Methods section. It is very long and difficult to follow.
- Smoking status in the follow-up assessment should probably be 7-day or 30-day point prevalence smoking abstinence instead “smoking every day or some days.” Smoking some days is ambiguous, especially if the inclusion criteria was smoking at least 1 cigarette in the past 30 days.
- In the Methods it is indicated that patient satisfaction was measured with one item using a Likert scale from 1 to 10. However, the results report qualitative data probably coming from open ended questions (page 11, lines 422-425).
- I do not think the first two sentences of the results need to be provided (page 7, lines 331-333).
- The results indicate that of the 984 patients with tobacco use information available in the EHR, 34.4% were “former smokers.” Since patients were still eligible to participate in the study if they had quit smoking within the past 30 days, I highly suggest being specific of what type of former smokers were these 34.4% (page 7, line 341).
- It seems that many patients were not approached because they did not have in person appointments or due to schedule conflicts. Could remote assessment be an option in the future to increase recruitment rates? (page 7, lines 344-348).
- Three patients were not approached following nurse’s recommendation (page 7, line 347). Could the authors be more specific on why the nurse discourage approaching these patients?
- From the 24 patients who showed interest in the study, the researcher was only able to reach 13 over the phone. I do not understand why these patients were reached over the phone. My understanding was that screening, consent, and baseline questionnaire were happening in person.
- The first paragraph of page 9 is confusing since differences were not significant. I recommend deleting the paragraph and just keep the last sentence to avoid confusion.
- Patients in the sample only had lung and head and neck cancers. Were patients with other cancer types not invited to participate? Please consider specifying this in the title.
- I agree with the authors on that EHR-based referral is not an option for some cancer centers. However, I do not believe it requires more resources than the patient navigator program presented in this study. The authors are comparing smoking cessation services (e.g., tobacco treatment specialist) with patient referral with the navigators, which still requires coordinators to identify patients who smoke and tobacco treatment specialists or other professionals to deliver smoking cessation treatment.
- Lack of differences between patients interested vs not interested in participating in the study may be due to the fact that all patients had a smoking related cancer. You should also consider comparisons by cancer stage and time since diagnosis, since those were predictors in previous studies.
Author Response
Dear Reviewer,
Thank you for your review. We appreciate the time and effort that you dedicated to providing feedback on our manuscript. We have incorporated most of the suggestions, please see attached document, in red, for a point-by-point response to your comments and concerns.
Thanks,
Tongyao

Reviewer 2 Report
Dear Authors,
There are never too many smoking cessation interventions. The studies on this topic are valuable because each of them brings something new regarding the possibility of reducing the harms resulting from exposure to tobacco smoke. Advice on the harmfulness of smoking and the possibility of stopping smoking should be part of every medical visit. In many countries, the time of GP or specialist out patient department is too short to provide a complete education on the subject. Therefore, the idea of ​​patient navigation program and its potential effectiveness described in this paper seems very interesting.
Author Response
Dear reviewer,
Thank you for reviewing. We really appreciate the positive comments.
Best,
Tongyao
Reviewer 3 Report
The introduction correctly points out than cancer patients who continue to smoke face greater risks than those who discontinue smoking. This paper incorrectly concludes that adding a patient navigation process to recruit cancer patient smokers to the use of FDA approved smoking cessation pharmaceuticals is both feasible and effective.
Their data do not support such their conclusion. As seen by this reviewer, the data presented show that patient navigation, as presented in this paper, is of little or no practical value.
Addressing feasibility requires consideration of the dollar costs of the patient navigation program, the extent to which these dollars can be recouped through health insurance billing, and the benefits and burdens of the program as seen by the patients, doctors, nurses, and health facility. None of these issues were addressed.
They screened 1,168 electronic medical records to identify what they considered to be 92 current smokers with a cancer diagnosis. They then approached 67 of them, 43 of which were not interested in participating in the navigation process. Of the 24 who were interested, only eight enrolled and completed two-month follow-up. Of the eight, six initiated use of one FDA approved smoking cessation medication for at least one day during the study period. Apparently, authors considered that 6 of these eight respondents as demonstration of the feasibility and efficacy of the patient navigation process.
This is nonsense. Six of 1,168 should have been the calculation for efficacy. In other words, 0.5% of those screened might have secured some benefit from this program. The standard of one day’s use of one medication during the study period was a ridiculously low standard of efficacy. It is also nonsense because there is no control population to rule out the possibility that, during the study period, the same number of cancer patients might have made yet another attempt to quit smoking based on the recommendations of their physicians.
Other data presented within the paper cast further doubt upon the navigation process, the author’s understanding of the efficacy of the FDA approved smoking cessation medications and the dynamics by which patients who quit smoking often relapse. Table 3 focusses in on the ten patients who initially enrolled in the navigation process. All had tried nicotine replacement therapy (NRT) products in the past, with only one having stayed quit for more than six months. Yet, the thrust of the navigation program was to get these patients to again try the same medications that have failed them in the past.
There is no discussion in the paper as to why so many of the patients had no interest in the navigation process, and why, of the 24 who did, why only 10 initiated the process. From the perspective of this reviewer, the most likely reason was that having failed in the past, most smoking cancer patients had no interest in trying these same medications again.
Yet another flaw in this study is the author’s lack of understanding of the differences between “tobacco use” and “smoking,” apparently considering these terms to be synonymous. They are not. Also, the title of the paper is in error. Their intervention was intended to secure smoking cessation, not tobacco harm reduction – again, two very different end points.
The only way I can envision this paper being revised in a way that would make it suitable for publication would be for the data presented to be clarified, reasons for non-participation and exclusion better considered, and use the data to reach the conclusion that patient navigation for purposes of enrolling cancer patient smokers in FDA approved smoking cessation regimens is a waste of time and money. Also, the title of the paper would need to be amended to reflect smoking cessation rather than tobacco harm reduction.
Again, it is important to state that cancer patients who continue to smoke face greater risks than those that discontinue smoking. To address this issue, the cancer center will have to find an intervention that will be attractive enough for patients to enroll and for physicians to endorse. Ideally, it would be effective enough to get patients uninterested in quitting to permanently quit. Until then, there is no point in pursuing the patient navigation process evaluated in this paper.
Some believe that such an intervention already exists, but what it is and how it might be implemented is beyond the scope of this review.
Author Response

(The authors gave the same response as above.)

Round 2
Reviewer 1 Report
I appreciate the authors’ being responsive to all my comments, however, I still have some additional concerns:
- I do not agree with the modification made to the study aim. I believe now it is not specific and does not match with the results and conclusion. This study is seeking to assess feasibility and acceptability of using patient navigation to connect cancer patients with smoking cessation barriers, and this should be clear.
- My major concern with this study is that the conclusion is not supported with the data. Only 35.8% of patients approached were interested in participating and only 12 patients ended up participating. This is a very low number and you cannot conclude that this is feasible. I did see that authors provided references of studies with similar recruitment rates, but this is not a good justification. Also, recruitment rates were higher in two of the studies referenced (Martinez et al., 2003; Japuntich et al., 2016) and only LeLaurin et al. 2020 had 16% of patients accepting to be in the study. The low recruitment rates reported in this study indicates that implementation is not feasible as the study is right now and that modifications need to be made to study procedures to increase accrual rates. This is why feasibility studies are conducted.
- Since this study addresses cigarette smoking, I do not think you need to introduce smokeless tobacco in the Discussion (first paragraph). I highly suggest removing this new addition.
- The new sentence added addressing having research staff involved in the process of identifying and assessing patients is out of place (p. 14, lines 525-527). I believe it would be better placed in the study limitations.
- The new paragraph regarding enrolling patients over the phone is not very clear. My understanding is that the authors would like to express that phone approaches have the limitations of losing connection. However, please pay attention that one thing is screening and assessing over the phone and another thing is conducting treatment over the phone. Remote approaches and assessments could be used only for those cases in which patients do not have in-person appointments, so they do not miss the opportunity of participating in the study. In addition, this contradicts the present study in which the navigators were connecting with patients over the phone.
Reviewer 3 Report
Please see the attachment
